# Effect of Sugar Beet Pulp and Anionic Salts on Metabolic Status and Mineral Homeostasis during the Peri-Parturient Period of Dairy Sheep

**DOI:** 10.3390/ani13020213

**Published:** 2023-01-06

**Authors:** Christina Peleki, Evangelos Kiosis, Zoe S. Polizopoulou, Georgios Tsousis, George C. Fthenakis, Nektarios D. Giadinis, Christos Brozos

**Affiliations:** 1School of Veterinary Medicine, Aristotle University of Thessaloniki, 54627 Thessaloniki, Greece; 2Veterinary Faculty, University of Thessaly, P.O. Box 199, 43100 Karditsa, Greece

**Keywords:** dairy sheep, macrominerals, sugar beet pulp, anionic salts

## Abstract

**Simple Summary:**

During the peri-parturient period, the development of the fetus and the production of the colostrum and milk lead to increased requirements of energy and minerals, which, in turn, predispose ruminants to the risk of metabolic diseases, such as hypocalcemia. Feed additives could provoke or prevent hypocalcemia. The aim of the study was to investigate the effect of sugar beet pulp administration during the dry period, alone or together with anionic salts, on the serum concentrations of calcium, magnesium, phosphate, and potassium, during the peri-parturient period of Chios dairy sheep. The results revealed a benefi cial effect of sugar beet pulp and anionic salts administration during the dry period in preventing peri-parturient hypocalcemia.

**Abstract:**

Sugar beet pulp is a popular by-product of sugar extraction; however, it can potentially cause depletion of Ca availability due to its oxalic content. The experiment examined the effect of sugar beet pulp and anionic salts administration during the dry period on the serum concentration of calcium, magnesium, phosphate, and potassium of dairy sheep. Eighty-seven sheep were divided into three groups (A, B, and C) according to their body condition score (BCS) and age at 40 days before the expected lambing. All groups received alfalfa hay, mixed grass straw, and a concentrate supplement. The concentrate fed to groups B and C contained sugar beet pulp. The nutritional value fed to all three groups was similar, except for Dietary Cation Anion Difference (DCAD). Animals of group A had a DCAD of +198 mEq/kg, animals of group B of +188 mEq/kg, and animals of group C were fed 20 gr/d ammonium chloride to achieve a negative DCAD (−52 mEq/kg). All groups were fed the same ration after lambing. Blood samples were collected 30 d, 20 d, 17 d, 14 d, 10 d, 7 d, and 4 d before lambing (a.p.), 6 h, 12 h, 24 h, 7 d, 10 d, and 15 d after lambing (p.p) for calcium, magnesium, phosphate, and potassium, and 30 d a.p., 7 d, and 15 d p.p. for beta hydroxybutyrate acid (BHBA) concentrations. Urine samples were also collected 20 d, 10 d, 4 d a.p., and 7 d p.p for the evaluation of pH levels. Ca levels of the control group decreased earlier and were lower at 4 d a.p. compared to those of group B and C. Additionally, the control group showed lower *p* values compared to group C at 20 d and 17 d a.p. P levels recovered earlier post parturition in young (age 1–1.5 years old) compared to older ewes. Group C had lower urine pH values throughout the pre-parturient period, reflecting the acidifying effect of the administered ammonium chloride, without any side effect on macromineral blood concentration. Feeding sugar beet pulp and systemic acidifying before parturition is considered safe and might even be beneficial in preventing hypocalcemia.

## 1. Introduction

The peri-parturient (or transition) period in dairy sheep is typically defined as the time frame from 3 weeks before to 3 weeks after lambing and is considered as one of the most critical periods for health and production. During this period, the demand for energy and other nutrients, such as minerals, increases rapidly to meet the requirements for fetus development, the colostrum and milk production [1,2]. At the same time, the rapidly expanding uterus limits the volume of the rumen and leads to decreased dry matter intake (DMI). Under these conditions, immense physiological changes take place to sustain homeostasis [3,4].

In dairy cows, DMI decreases dramatically during the last 2–3 days of pre-calving and takes several days after parturition to recover [5,6]. Nowadays, due to genetic selection and improved management, dairy ewes yield more milk. Therefore, decreased food intake around parturition imposes the animal to the risk of metabolic diseases, such as pregnancy toxemia, ketosis [7,8,9], hypocalcemia, and hypomagnesemia [10]. β-hydroxybutyric acid (BHBA) concentrations have been successfully proposed as indicators for the diagnosis of pregnancy toxemia and ketosis [11].

Hypocalcemia is a common metabolic disease in dairy cows caused by decreased levels of serum calcium (Ca) [12,13,14]. However, in contrast to dairy cows, where disease is noticed shortly before or after calving, hypocalcemia in dairy ewes can develop from several weeks before lambing to the first 2 weeks post-lambing [10]. While in dairy cows the most important cause is the rapid increase in milk yield and concomitant Ca demand, the mechanism of hypocalcemia in ewes might involve dietary imbalances [10].

Sugar beet pulp is a by-product of sugar extraction and is considered a highly digestible, palatable fiber source; therefore, it is a popular feed for ruminants [15,16]. Sugar beet pulp contains 0.89 units for lactation (UFL), 8.1% Crude Protein (CP), 20.6% Acid Detergent Fiber (ADF), 40.5% Neutral Detergent Fiber (NDF), 2.7% Ca, and 0.8% P, and is deficient in fat, carotene, and certain B complex vitamins [15]. It contains between 3.3 and 4.9% dry matter (DM) oxalic acid, which can rapidly form non absorbable salts with Ca [17] and Mg [18], resulting in a sudden depletion of these minerals. High oxalate, potassium, and crude protein levels in sugar beet pulp play an important role in the occurrence of hypocalcemia and hypomagnesemia [19].

A dietary strategy of decreased Dietary Cation Anion Difference (DCAD) fed pre-partum has been extensively investigated in recent years, especially in cows, with the assumption that acidifying the diet can allow high calcium regimes to be fed during the late dry period, without causing hypocalcemia [20,21,22]. Recent studies [23] indicated that feeding a low DCAD diet to late pregnant ewes should be accompanied with Ca supplementation. Sugar beet pulp has been previously fed to late pregnant ewes and proved beneficial for their productivity [24], but its effects on macromineral homeostasis has not been investigated. However, studies investigating the effect of Ca binding nutrients (i.e., beet pulp) in combination with the use of decreased DCAD as a preventative measure in diary ewes are lacking in the literature.

The aim of this study was to investigate whether (a) including sugar beet pulp in the concentrate during the last month of gestation could lead to the imbalance of the homeostatic mechanism of macrominerals and contribute to the appearance of metabolic disorders and (b) to investigate the ability of the anionic salts to prevent the onset of this potential predicament.

## 2. Materials and Methods

### 2.1. Animals and Feeding Regimes

The experimental group consisted of eighty-seven Chios ewes and yearlings, housed indoors in the premises of Aristotle University of Thessaloniki, Veterinary Medicine Department of Farm Animals, Kolhiko, Greece. All animals were monitored for health status; during their enrollment to the study, no signs of disease were found in any of the animals. They were equally allocated into three experimental groups based on their age and body condition score (BCS). The animals were previously synchronized with intravaginal sponges, inseminated on the same day, and lambed 149 ± 7 d after insemination.

Diets were formulated to meet the nutritional requirements of pregnant sheep (Institute Nationale de la Recherche Agronomique (INRA, 1989)) except for sodium, potassium, chloride, sulfate, and phosphate. These minerals met or exceeded the requirements of pregnant sheep dependent upon the target cation–anion difference of the ration. The DCAD was calculated using the equation DCAD (mEq/kg of DM) (K × 256 + Na × 434)—(Cl × 282 + S × 624) [25]. The sheep of each experimental group received their respective rations beginning 40 d pre-partum until lambing. Dietary components were offered as pellets twice daily.

Ewes received alfalfa hay (1 kg/animal/day), mixed grass straw (ad libitum), and concentrated mixture (800 g/animal/day). Specifically, the pellet mixture of group A (control group) contained corn, barley, soya, wheat bran, vitamins, and minerals; the pellet mixture of group B contained corn, soya, beet pulp, vitamins, and minerals; the pellet mixture of group C contained the same nutrients as group B plus 25 gr of ammonium chloride (Table 1). All three concentrates were similar in major nutritional values with regards to DM (Group A: 0.94 UFL, 13.5% CP, 3.1% Fat, 19.5% NDF, 1.11% Ca, 0.35% Mg, 0.28% P, 1.87 K, 0.21 Na, 0.77 Cl, 0.24 S; Group B: 0.95 UFL, 11.5 CP, 2.1% Fat, 22.5 NDF, 1.11% Ca, 0.35 Mg, 0.27 P, 1.74 K, 0.2 Na, 0.66 Cl, 0.26 S; Group C: 0.95 UFL, 13.3 CP, 2.0%, 22.8 NDF, 1.09 Ca, 0.37 Mg, 0.30 P, 1.59 K, 0.21 Na, 1.40 Cl, 0.25 S) and differed in DCAD (Group A: +198; Group B: +188; Group C: −52 mEq/kg of DM). During the experimental period before lambing, the groups were housed separately, while after parturition they formed one group and were fed with the same ration to meet the requirements for daily milk production of 2 kg.

Feed samples (forages and pellets) were analyzed by the Independent Testing Laboratory VitaTrace Nutrition Ltd., Cyprus. The analysis for minerals was performed by the Diagnostic Laboratory of Veterinary Medicine of Aristotle University of Thessaloniki (AUTH).

### 2.2. Blood Sampling

Blood samples were obtained by jugular vein puncture at 30 d, 20 d, 17 d, 14 d, 10 d, 7 d, and 4 d antepartum (a.p.), and 6 h, 12 h, 24 h, 7 d, and 10 d postpartum (p.p.), with the ewes being lightly restrained. The blood was collected directly into vacutainer tubes (BD Vacuntainer^®^, 10 mL, Plymouth, UK) using an 18-gauge needle and a holder. The samples in the tubes were allowed to clot at room temperature for 30 min and were centrifuged in 1600× *g* for 15 min to separate serum. After coagulation, serum was transferred in plain tubes and stored at −20 °C until analysis at the Diagnostic Laboratory, School of Veterinary Medicine, Faculty of Health Sciences, Aristotle University of Thessaloniki, Greece.

The values of BHBA were measured on-site 4 h after feeding at three time points: d30 a.p., d7 p.p., and d15 p.p. using a hand-held Freestyle Precision Xceed meter (Abbott, Abbott Diabetes Care Ltd., Oxon, UK) with single use ketone strips [11]. The measurement was carried out within 15 min of sampling following the manufacturers’ instructions and at room temperature between 21 and 23 °C.

### 2.3. Urine Sampling

The urine samples were collected after free urination at four time points: 20 d, 10 d, 4 d a.p., and 7 d p.p. Urine pH was determined using a standardized pH meter (AD12 Waterproof pH Tester, Adwa, Szeged, Hungrary). The use of a glass electrode pH meter was the reference method for measuring urine pH [26].

### 2.4. Biochemical Analyses

The serum concentrations of Ca and P were determined by a clinical chemistry analyzer (Flexor F, Vital Scientific, Spankeren, The Netherlands). The evaluation of serum concentrations of K, Na, and Mg was measured with the aid of flame atomic emission spectrophotometry (Analyst 100 PerkinElmer; Bodenseewerk PerkinElmer Gmbh, Postfach 101761, D-88647, Ueberlingen, Germany).

### 2.5. Ethical Statement

The study was performed during the lambing season (October–November) on a single farm of Aristotle University of Thessaloniki in Kolhiko. The study protocol was reviewed by the Committee of Ethics in Animal Research of Thessaloniki (Permission number: 98864/659). All animals were handled in accordance with acceptable practices and experimental protocols reviewed and approved by the European Union Directive 2010/63/UE concerning the protection of animals used for research.

### 2.6. Statistical Analysis

To investigate the effect of the dry period diet or age of the ewe on the levels of a series of biomarkers (blood concentration of Ca, Mg, P, K, urine pH, and β-hydroxybutyric acid value), while as well as considering sampling time, linear mixed-effect models for repeated measures were fitted, using the levels of each biomarker as a dependent variable. Τhe nutritional and age group were fitted separately in the models, which also included sampling time and their pairwise interactions as independent variables. The age group was not utilized in the models of urine pH and β-hydroxybutyric acid. The ID of each animal was used as a random effect.

Post-hoc analyses focused on interactions of interest and were carried with the emmeans R package for estimating unbiased marginal means for our mixed effects models. Their respective graphical displays were included, and pairwise test values, in a robust way, were based on the model’s predictions. The significance level was set at *p* < 0.05.

## 3. Results

Regarding Ca levels of the three nutritional groups, mild differences between the groups were noticed up to day 7 a.p. Specifically, group A had numerically lower values during most of this period, except for day 17 a.p., where a significant difference between this group compared to group B was detected (*p* = 0.05, Figure 1) From day 4 a.p., a drastic drop in Ca levels of group A was noticed, whereas this drop was found at 6 h p.p. in the other two groups. As a result, Ca levels of group A differed significantly from those of groups B and C at 4 d a.p. (*p* = 0.03 and *p* = 0.005, resp.). Ca levels reached a nadir in all groups at 12 h p.p. and recovered thereafter in a uniform manner in all groups.

Similar observations were noticed for the Ca levels of the three age groups across time. In all groups, a sharp drop in blood Ca from 4 d a.p. and a sharp increase from 12 h p.p. was noticed (Figure 2). Ewes from 2–2.5 years old showed higher Ca levels compared to the age group 3+ at 14 d a.p. (*p* = 0.005) and compared to group 1–1.5 at 4 d a.p. (*p* = 0.02). At 6 h p.p., Ca levels of age group 3+ were higher compared to those of group 1–1.5 (*p* = 0.04), with group 2–2.5 showing intermediate levels.

Regarding Mg levels, milder fluctuations were noticed in all nutritional groups across our study (Figure 3). In all groups, a non-significant drop was noticed around parturition and values recovered slowly until the end of the experimental period. No statistical differences were detected between nutritional groups at any time point. Ewes 3+ years old showed lower levels of Mg compared to group 2–2.5 at 14 d a.p. (*p* = 0.01) and compared to group 1–1.5 at 7 d a.p. and at 4 d a.p. (*p* = 0.001 and *p* = 0.04, resp., Figure 4)

The same trend of peri-parturient drop was evident as well in P levels (Figure 5), with values in every group showing larger fluctuations compared to Ca and Mg values. Animals receiving diet C had increased levels of P compared to animals with diet A at 20 d a.p. (*p* = 0.05) and at 17 d a.p. (*p* = 0.02), with ewes in group B showing intermediate values.

Ewes from 1–1.5 years old recovered P levels sooner, already from 12 h p.p., compared to the other two age groups that recovered after the first 24 h p.p. (Figure 6). As a result, group 1–1.5 showed higher P levels compared to group 2–2.5 at 12 h p.p. (*p* = 0.03) and compared to both groups 2–2.5 and 3+ at 24 h p.p. (*p* = 0.005 and *p* = 0.03, resp.).

Potassium levels showed a more gradual decrease as ewes approached parturition and increased afterwards (Figure 7). Differences between the groups were only evident at the initiation of the experiment. Moreover, no differences whatsoever were found between age groups (Figure 8).

The urine pH remained lower throughout the pre-parturient period (day 20, 10, and 4 a.p.) in the nutritional group fed with the anionic diet (group C) compared to the other two groups (Figure 9), confirming the expected action of ammonium chloride in systemic acidification. After parturition, where diets were unified, no differences between groups were detected.

Regarding BHBA, blood levels increased gradually from 30 d a.p. to 15 d p.p. in all groups in a uniform manner. However, no statistical differences were detected (Figure 10).

## 4. Discussion

The aim of the study was to investigate the effect of sugar beet pulp with or without anionic salts on the serum concentrations of calcium, magnesium, phosphate, and potassium during the peri-parturient period of Chios dairy sheep.

Peri-parturient hypocalcemia is the most common metabolic disease in dairy ruminants. It predisposes to dystocia, uterine prolapse, mastitis, retained placenta, ketosis, and decreased rumen and abomasum motility [27,28,29,30]. Yildiz et al. [2] reported a significant decrease in the serum levels of Ca in ewes when close to parturition. Beet pulp can potentially lead to Ca deficiency due to contained oxalic acid, which forms non-absorbed salts with dietary calcium in the rumen [18,19]. However, Kohestani et al. [24] found beneficial effects after beet pulp administration in late pregnancy, but the amount fed in this trial was lower compared to the one in our study.

Prevention of clinical and subclinical hypocalcemia can be achieved by activating the regulatory mechanisms of Ca homeostasis during the last days of gestation. Negative Dietary Cation Anion Difference (DCAD) between −50 and −150 mEq/kg of DM during the last 10 to 28 d of gestation is proposed to promote Ca homeostatic mechanisms to facilitate blood Ca concentrations within normal levels in dairy cows [31,32]. This theory is based on the development of metabolic acidosis to enhance tissue responsiveness to parathyroid hormone (PTH) [33] and, thus, increase the available calcium in blood [34]. Parathyroid hormone stimulates the mobilization of Ca from the bones and decreases renal Ca excretion. Furthermore, PTH enhances the conversion of 25-dihydroxyvitamin D3 to 1.25- dihydroxyvitamin D_3_, which stimulates the gastrointestinal absorption of Ca [35]. A widely used acidogenic agent to decrease DCAD in livestock is ammonium chloride. Indeed, the results of our study indicated that animals fed with negative DCAD (group C, DCAD = −52 mEq/kg of DM) showed a slower decrease in Ca concentration a.p. compared with the control (Group A, DCAD = +198 mEq/kg of DM). Interestingly, the same effect was noticed for animals fed sugar beet pulp without ammonium chloride (Group B, DCAD = +188 mEq/kg of DM), implying that anionic diets are not the only regulating mechanism for Ca homeostasis. Urine pH provides an inexpensive and practical method monitoring the degree of systemic acidification [36]. The animals in group C had significantly lower pH levels in comparison with group A and group B, confirming the acidification caused by the ammonium chloride. However, the values in animals with a decreased DCAD diet remained within the rational range without causing any extreme acidosis [37].

Normal blood calcium levels in Chios sheep are from 2.18–2.75 mmol/L [38]. The results from the present study indicate low blood Ca levels in all groups from 4 d a.p. to 12 h p.p. (Figure 1), which is in accordance with previous studies [39]. This decrease in serum Ca concentrations was previously attributed to the reduced DMI around parturition [1,20,40] and the increased demand for Ca for colostrum formation, milk production [20,41,42], and fetal skeleton formation [43]. Interestingly, animals in Group A showed an earlier drop in Ca concentration compared to the other two groups. This leads to the assumption that the sugar beet pulp fed alone (Group B) or in combination with the acidogenic agent (Group C) contributed to a less acute drop in peri-parturient blood Ca concentration, probably due to enhanced utilization of food components through the improvement of palatability. This was also previously reported by Charbonneau et al. [31] in dairy cows. Age has been reported to be a major factor affecting blood Ca concentration in the peri-parturient period [44,45], which was in line with the results from our study (Figure 2). More specifically, older animals (3+ years old) showed a smoother drop in Ca levels just after lambing (6 h p.p.), which could be attributed to higher resilience. Additionally, intermediate age ewes (2–2.5 years old) seem to sustain more elevated levels of Ca before parturition.

Magnesium is a significant cation located both in intracellular and extracellular fluid. Normal magnesium levels in blood serum of Chios sheep range between 0.94 and 1.31 mmol/l [38]. Despite its multiple and essential functions, Mg homeostasis is not regulated by a hormonally controlled feedback system. The principal mechanisms of maintaining Mg within the normal range is the absorption from the gastrointestinal track and the excretion via urine based on the animal’s requirements [46]. Magnesium requirements raise significantly in late pregnancy and early lactation, putting the animal at the risk of subclinical or clinical hypomagnesemia [10]. Mobilization of Mg from bone is ineffective because the ratio of Ca:Mg is 42:1 and it would disrupt Ca homeostasis [47]. Lactation and late gestation can result in large losses of Mg from intracellular and extracellular fluid [48], which was also evident in our study, especially from 6 h to 12 h p.p. when udder padding with milk commenced (Figure 3). Plasma Mg^2+^ is known to be influenced in a non-specific manner by catecholamines [49], insulin [50], and parathyroid hormone (PTH) [34]. The results from our research revealed no differences among nutritional groups throughout the experimental period. In contrast, Yildiz et al. [2] observed a significant decrease in Mg serum levels during the pre-partum period in ewes. Additionally, age did not seem to significantly influence Mg concentration, although older animals (+3 years old) seemed to show lower levels of Mg before lambing compared to young animals and this could be attributed to a weaker hormonal response to gradually increased demands as parturition approached.

Normal phosphate blood levels range between 1.62 and 3.33 mmol/L [38]. Stress of parturition is followed by an elevation of cortisol levels in blood serum, which is negatively correlated with the serum levels of Ca and P, resulting in hypophosphatemia [21]. Results from the present study indicate significant differences in blood P concentrations during the peri-parturient period, which agrees with earlier findings [51]. Nevertheless, despite the more evident fluctuations among nutritional groups compared with Ca and Mg, no remarkable differences were noticed. Intriguingly, young ewes (1–1.5 years old) showed a prompter recovery of P concentrations during the first day after lambing compared with the older ones. A possible explanation might be the different P needs in young animals, a hypothesis which needs further investigation. Feces and milk are the most important routes for phosphate excretion in ruminants [52]. The ruminant mammary gland maintains constant concentrations of Ca and P in milk, independently of the availability of these elements to the maternal organism. It is possible that multiparous ewes, through their higher potential for milk production, tended to withdraw increased amounts of P, which delayed its recovery in blood serum.

Hypophosphatemia and variations in levels of serum Mg also occur concurrently with hypokalemia. The latter is a metabolic disorder associated with Ca homeostasis that affects approximately between 2% and 6% of postpartum ewes [44]. All four elements, Ca, P, Mg, and K are closely related to many metabolic events in the body. Potassium, however, is not stored to any large extent in the bone, it exists in the body primarily as a cellular constituent. As the animal ages, the readiness of availability of stored mineral elements in the bone decreases [53,54]. A high potassium concentration in the diet is a risk factor for hypomagnesemia as it reduces Mg absorption from the rumen and lowers Ca levels [55]. In our study, K levels showed a gradual decrease from the beginning of the transition period until lambing to recover afterwards in all nutritional or age groups. This pattern is in accordance with findings in the other measured macrominerals, reflecting their interrelation during the peri-parturient period.

The determination of BHBA blood concentrations revealed no significant differences between the three nutritional groups and remained within the normal range throughout the examination period [56,57]. Blood BHBA concentration was previously reported to increase during late gestation and may reach its peak before or around lambing [9,58,59]. Nevertheless, elevated values of BHBA found in individual animals of all three groups (data not shown) raise concerns about the guidelines of INRA for Chios sheep and, therefore, need further investigation.

## 5. Conclusions

The hypothesis that feeding sugar beet pulp to dairy sheep before parturition could lead to hypocalcemia was rejected based on the findings of our study. On the contrary, it proved to have a beneficial effect in preventing hypocalcemia in the fed quantities, regardless of the metabolic acidosis caused when adding ammonium chloride. This is possibly a consequence of a better utilization of food components through the improvement of palatability.

## Figures and Tables

**Figure 1 animals-13-00213-f001:**
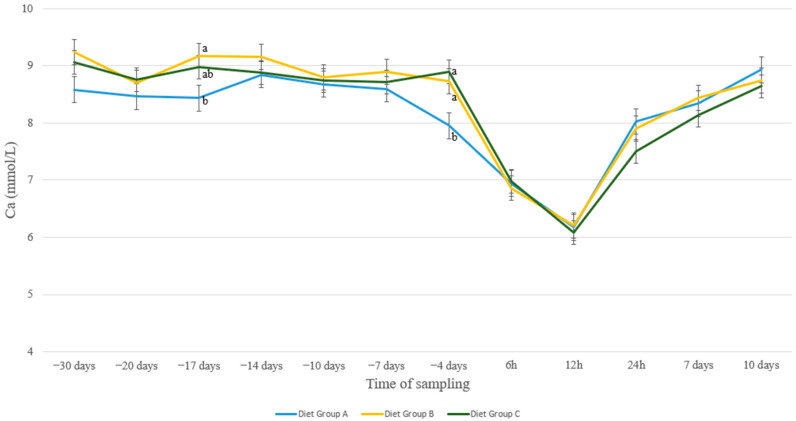
Ca levels (mmol/L) in blood serum of sheep in group A (control group, *n* = 27), group B (supplemented diet with beet pulp, *n* = 31), and group C (supplemented diet with beet pulp and ammonium chloride, *n* = 29) during the peri-parturient period. ^a,b^ Different letters denote significant differences between groups within evaluation time (*p* < 0.05).

**Figure 2 animals-13-00213-f002:**
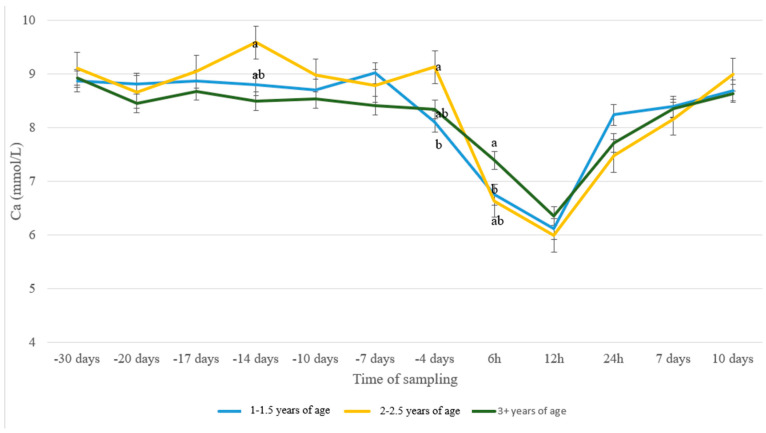
Ca levels (mmol/L) in blood serum of sheep in age groups 1–1.5 (*n* = 35), 2–2.5 (*n* = 11) and 3+ (*n* = 41) years old during the peri-parturient period. ^a,b^ Different letters denote significant differences between groups within evaluation time (*p* < 0.05).

**Figure 3 animals-13-00213-f003:**
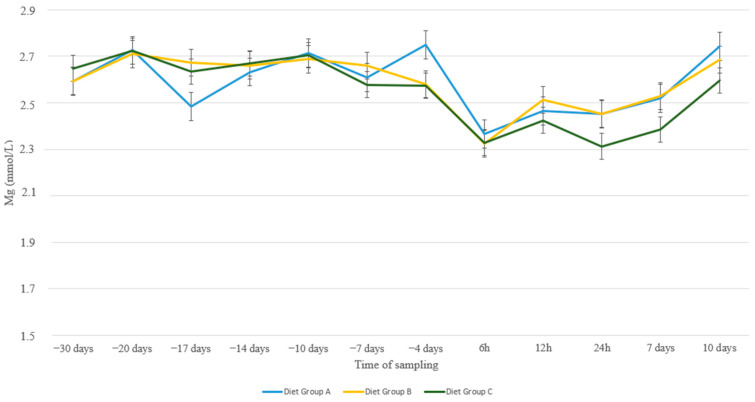
Mg levels (mmol/L) in blood serum of sheep in group A (control group, *n* = 27), group B (supplemented diet with beet pulp, *n* = 31), and group C (supplemented diet with beet pulp and anionic salts, *n* = 29) during the peri-parturient period.

**Figure 4 animals-13-00213-f004:**
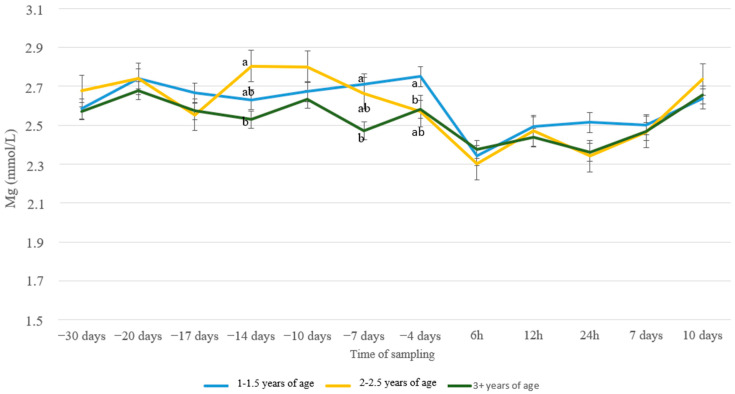
Mg levels (mmol/L) in blood serum of sheep in age groups 1–1.5 (*n* = 35), 2–2.5 (*n* = 11) and 3+ (*n* = 41) years old during the peri-parturient period. ^a,b^ Different letters denote significant differences between groups within evaluation time (*p* < 0.05).

**Figure 5 animals-13-00213-f005:**
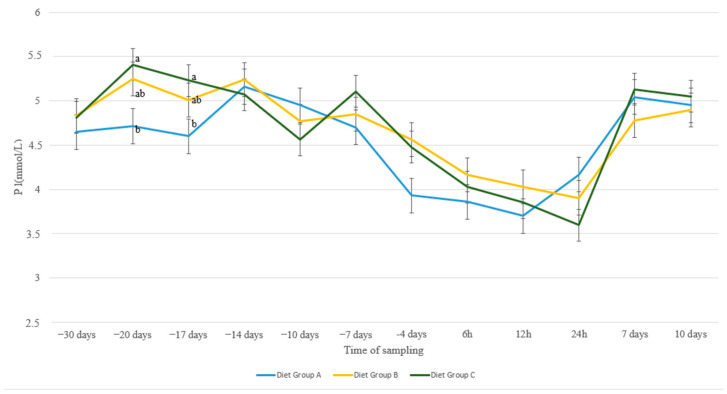
P levels (mmol/L) in blood serum of sheep in group A (control group, *n* = 27), group B (supplemented diet with beet pulp, *n* = 31), and group C (supplemented diet with beet pulp and ammonium chloride, *n* = 29) during the peri-parturient period. ^a,b^ Different letters denote significant differences between groups within evaluation time (*p* < 0.05).

**Figure 6 animals-13-00213-f006:**
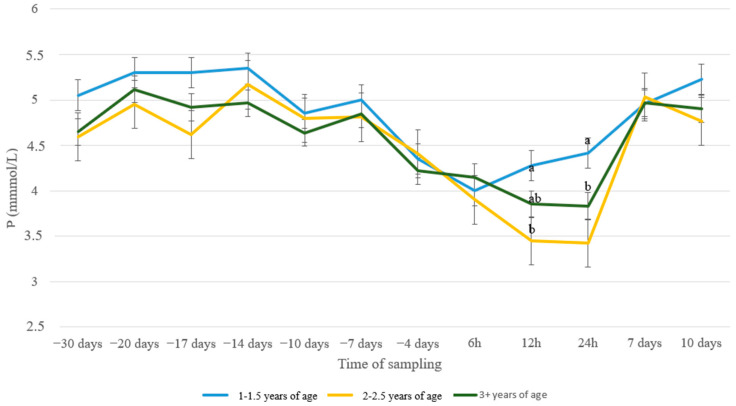
P levels (mmol/L) in blood serum of sheep in age groups 1–1.5 (*n* = 35), 2–2.5 (*n* = 11) and 3+ (*n* = 41) years old during the peri-parturient period. ^a,b^ Different letters denote significant differences between groups within evaluation time (*p* < 0.05).

**Figure 7 animals-13-00213-f007:**
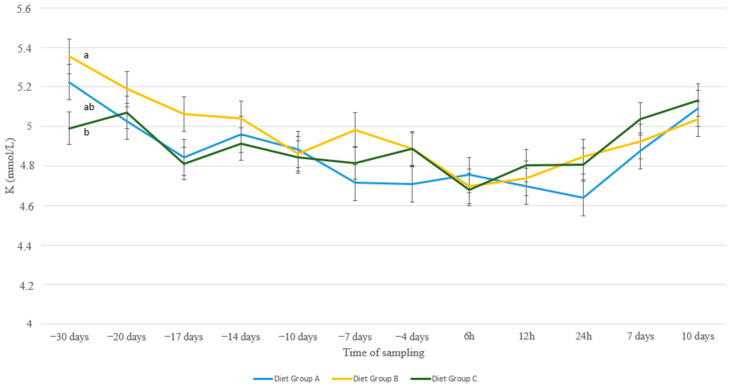
K levels (mmol/L) in blood serum of sheep in group A (control group, *n* = 27), group B (supplemented diet with beet pulp, *n* = 31), and group C (supplemented diet with beet pulp and ammonium chloride, *n* = 29) during the peri-parturient period. ^a,b^ Different letters denote significant differences between groups within evaluation time (*p* < 0.05).

**Figure 8 animals-13-00213-f008:**
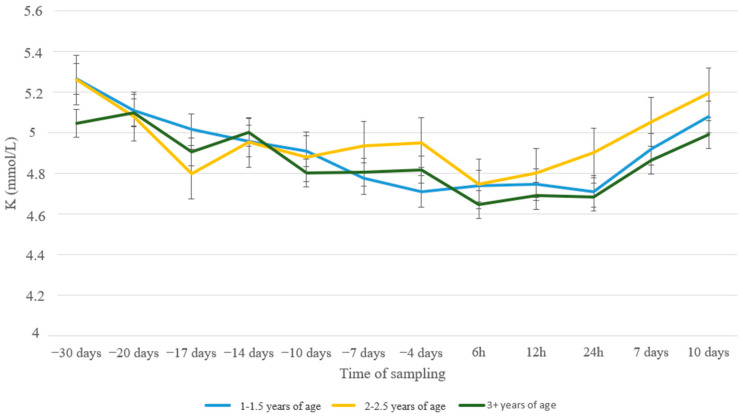
K levels (mmol/L) in blood serum of sheep in age groups 1–1.5 (*n* = 35), 2–2.5 (*n* = 11) and 3+ (*n* = 41) years old during the peri-parturient period.

**Figure 9 animals-13-00213-f009:**
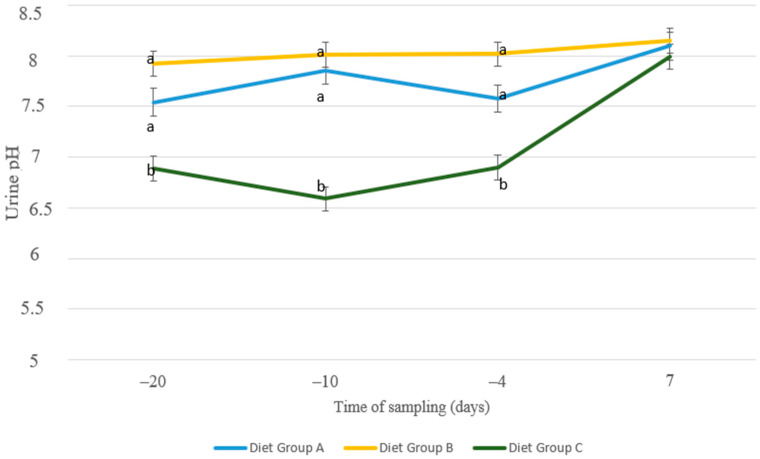
Urine pH of sheep in group A (control group, *n* = 27), group B (supplemented diet with beet pulp, *n* = 31), and group C (supplemented diet with beet pulp and ammonium chloride, *n* = 29) during the peri-parturient period. ^a,b^ Different letters denote significant differences between groups within evaluation time (*p* < 0.05).

**Figure 10 animals-13-00213-f010:**
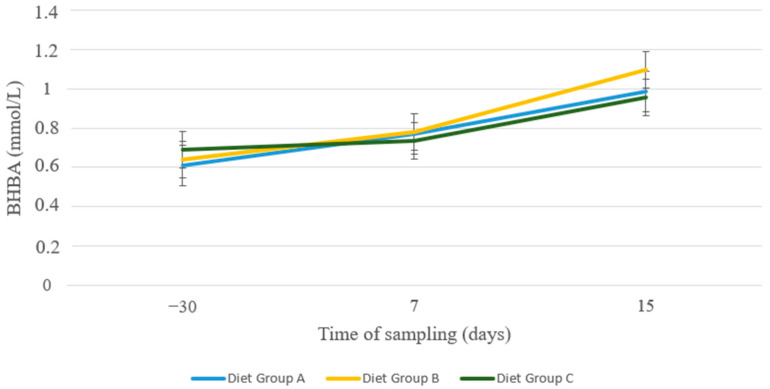
BHBA (mmol/L) in blood of sheep in group A (control group, *n* = 27), group B (supplemented diet with beet pulp, *n* = 31), and group C (supplemented diet with beet pulp and ammonium chloride, *n* = 29) during the peri-parturient period.

**Table 1 animals-13-00213-t001:** Composition of concentrate pellet mixture determined for every 1000 kg of food of each experimental group.

Concentrate	Group A	Group B	Group C
Corn	500	469	444
Barley	72	-	-
Soya	100	100	100
Wheat bran	290	-	-
Sugar beet pulp	-	400	400
NH_4_Cl	-	-	25
CMV *	2	2	2
CaCo_3_	18.5	-	-
CaHPO_4_	1.5	14	14
NaCl	10.5	7	7
MgO	5.5	8	8

***** CMV: Complex of minerals and vitamins.

## Data Availability

The data are available upon request from the corresponding author. The data are not publicly available as they form part of the Ph.D. thesis of the first author, which has not yet been examined, approved, and uploaded in the official depository of PhD theses from Greek Universities.

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
