# Peer review of "Effect of Sugar Beet Pulp and Anionic Salts on Metabolic Status and Mineral Homeostasis during the Peri-Parturient Period of Dairy Sheep"

_animals, 2023, doi:10.3390/ani13020213_

Round 1

Reviewer 1 Report

1. Introduction

Linhas 54 a 60

"Sugar beet pulp is a by-product of sugar extraction and is considered a highly digest-54 ible, palatable fiber source; therefore a popular feed for ruminants (Bhattacharya and Slei-55 man, 1971; Mustafa, 2011). Sugar beet pulp contains 9.1% CP, 31% ADF, 0.72% Ca and 56 0.2% P (Mustafa et. al., 2009) and is deficient in fat, carotene and certain B complex vita-57 mins (Bhattacharya et al., 1971). It contains oxalic acid 3.3-4.9% of DM which forms non 58 absorbable salts with Ca2+ (Bendary et al., 1992d). This depletion of dietary Ca can poten-59 tially lead to Ca deficiency".

Comment - this information suggests the real need to present the composition of the diets in the methodology

2. Materials and Methods

  Insert the diet composition table (dry matter, crude protein, neutral detergent fiber, detergent fiber, ether extract, ash, digestible or metabolizable energy, total digestible nutrients - Important questions to be clarified: Are diets isoenergetic? Are the diets isoproteic? - Present the composition of the mineral core; - Present the cation and anion balance   Lines 82 to 84 "Ewes received alfa alfa hay (1kg/animal/day), mixed grass straw (ad libitum) and concentrated mixture (800g/animal/day) to meet nutrient requirements (Institute Natio nale de la Recherche Agronomique (INRA, 1989) for 40 days before expected lambing."   Comment - inform in detail the formulation of the diet. What production level (Milk? Gain?) should the diet meet?   Lines 85 to 88

"Specifically, pellet mixture of group A (control group) contained corn, barley, soya, wheat 85 bran, vitamins, and minerals; pellet mixture of group B contained corn, soya, beet pulp, 86 vitamins and minerals and pellet mixture of group C contained the same nutrients as 87 group B plus 25 gr of ammonium chloride (Table 1)."

- Comment - informing the ingredients of the diet without presenting the composition generates doubts and confusion

4. Discussion

Lines 241 to 243

"Negative Dietary Cation Anion Balance (DCAD) is proposed to promote Ca homeostatic 241 mechanisms to facilitate blood Ca concentrations within normal levels (Charbonneau et 242 al., 2006; Lean et al., 2019)".

Comment - this discussion could be associated with diet composition (Calcium for example)

Reviewer 2 Report

Manuscript numer: animals-2114576

Tittle:  Effect of sugar beet pulp and anionic salts on metabolic status and mineral homeostasis during the peri-parturient period of dairy sheep

The Authors aimed to evaluate the effect of sugar beet pulp and anionic salts administration during the periparturient period on the serum concentration of calcium, magnesium, phosphate, and potassium of dairy sheep. It must be said that the research was planned very carefully. The methodology of the research was described in a transparent manner.

 The manuscript is well written, in fluent English language: clear, precise, and easy to understand. Also, it offers interesting insight on the need to expand the knowledge about factors affecting production traits in livestock.

In particular Authors found that feeding sugar beet pulp before parturition did not increase the risk of hypocalcemia. On the contrary, it proved to have a beneficial effect in preventing hypocalcemia in the fed quantities. This means that sugar beet pulp can be used as a component that improves the palatability of feed and thus better utilization of food components.

My general judgment is “Accept in present form”.

Author Response

We thank you very much for the positive evaluation of the manuscript.

Reviewer 3 Report

Animals-2114576 submitted by Peleki et al. “Effect of sugar beet pulp and anionic salts on metabolic status and mineral homeostasis during the peri-parturient period of dairy sheep” examined the potential of sugar beet pulp and anionic salts administration during the dry period on the serum concentration of calcium, magnesium, phosphate, and potassium using Eighty-seven dairy sheep divided into three groups for 40 days before the expected lambing. The authors concluded that feeding sugar beet pulp before parturition is considered safe and might even be beneficial in preventing hypocalcemia. Generally, a well written manuscript, relevant and interesting and important for the field. The article has scientific and practical impact however, the English needs to be revised along with quality of the manuscript should be improved. I think that this manuscript could be accepted in Animals but needs minor modifications and attentions before the publication. Some references are old. Can you confirm this with more recent references?.

Details of major comments to the manuscript are given below:

I´ll only highlight some general comments:

- I would recommend adding an introductory sentence at the start of the abstract.

- Check journal recommendations for References and P values, be consistence in this regard.

- Please write the full name of the abbreviations in their first mentions and throughout the manuscript.

- The introduction of manuscript is diluted and semi adequate in respect to reviewing the latest literature relative to the study described and should be improved somewhat.

- Please emphasis on the added value/novelty of this article as there are some researches utilizing such sugar beet pulp and anionic salts in the literature. Please provide more recent literature reviews and citations that can be related to the forms and way of application of sugar beet pulp and/or anionic salts.

- The hypothesis is not clear, and objectives in introduction does not completely fit to each other. Recheck it again. I would add a few sentences explaining what additional information your study will bring. As it is, it seems that your experiment is demonstrating something that has already been shown.

- The results section needs to be shortened with emphasis on important and economical findings.

- I suggest omitting all sentences with is no significant effect observed.

- I would suggest making every effort to explain more the results and not compare them with other studies. In general, the production data are interesting but did not discussed well.

- The discussion should include further arguments between studies or results found in the literature.

Author Response

Reviewer #3:

Animals-2114576 submitted by Peleki et al. “Effect of sugar beet pulp and anionic salts on metabolic status and mineral homeostasis during the peri-parturient period of dairy sheep” examined the potential of sugar beet pulp and anionic salts administration during the dry period on the serum concentration of calcium, magnesium, phosphate, and potassium using Eighty-seven dairy sheep divided into three groups for 40 days before the expected lambing. The authors concluded that feeding sugar beet pulp before parturition is considered safe and might even be beneficial in preventing hypocalcemia. Generally, a well written manuscript, relevant and interesting and important for the field. The article has scientific and practical impact however, the English needs to be revised along with quality of the manuscript should be improved. I think that this manuscript could be accepted in Animals but needs minor modifications and attentions before the publication. Some references are old. Can you confirm this with more recent references?

Response: Thank you very much for your kind words and motivation. The article has been further revised by a professional editor for more concise language.

Details of major comments to the manuscript are given below:

I´ll only highlight some general comments:

- I would recommend adding an introductory sentence at the start of the abstract.

Response: Thank you very much for your suggestion. We have added an introductory sentence in lines 18-19. Lines 18-19: Sugar beet pulp is a popular by-product of sugar extraction, however it can potentially cause depletion of Ca availability due to its oxalic content.

- Check journal recommendations for References and P values, be consistence in this regard.

Response: Thank you very much for pointing this out. We carefully checked and revised references and p values throughout the manuscript.

- Please write the full name of the abbreviations in their first mentions and throughout the manuscript.

Response: Thank you for your recommendation. We have corrected the abbreviations style throughout the manuscript.

- The introduction of manuscript is diluted and semi adequate in respect to reviewing the latest literature relative to the study described and should be improved somewhat.

Response: We very much appreciate reviewer’s suggestion and we have updated literature review with addition of recent publications [References 18,37,42].

[18] El-Khodery, S.; El-Boshy, M.; Gaafar, K.; Elmashad, A. Hypocalcemia in Ossimi Sheep Associated with Feeding on Beet Tops (Beta vulgaris). Turk J. Vet. Anim. Sci. 2008, 32, 199-205.

[37] Kohestani, M.G., Yansari, A.T.; Rezaei, M. Effects of partial replacement of barley with sugar beet pulp on pre-and post-partum performance of Zel ewes. South African Journal of Animal Science 2011, 41, 256-264.

[42] Masters, DG.; Hancock, S.; Refshauge, G.; Robertson, S.M.; McGrath, S.; Bhanugopan, M.; Friend, M.A.; Thomson, A.N. Mineral supplements improve the calcium status of pregnant ewes grazing vegetative cereals. Animal Production Science 2019, 59, 1299-1309. doi:10.1071/AN16530.

- Please emphasis on the added value/novelty of this article as there are some researches utilizing such sugar beet pulp and anionic salts in the literature. Please provide more recent literature reviews and citations that can be related to the forms and way of application of sugar beet pulp and/or anionic salts.

Response: As suggested by the reviewer we tried to explain more carefully in relation with the recent literature the novelty of our study. Added information are included in lines 71-81. Lines 71-81: A dietary strategy of decreased Dietary Cation Anion Difference (DCAD) fed pre-partum has been extensively investigated in recent years, especially in cows, with the assumption that acidifying the diet can allow high calcium regimes to be fed during the late dry period, without causing hypocalcemia [26,29,33]. Recent studies [42] indicated that feeding low DCAD diet in late pregnant ewes should be accompanied with a Ca supplementation. Sugar beet pulp has been previously fed in late pregnant ewes and proved beneficial for their productivity [37], but its effects on macromineral homeostasis has not been investigated. However, studies investigating the effect of Ca binding nutrients (i.e. beet pulp) in combination with the use of decreased DCAD as a preventing measure in diary ewes are lacking in literature.

- The hypothesis is not clear, and objectives in introduction does not completely fit to each other. Recheck it again. I would add a few sentences explaining what additional information your study will bring. As it is, it seems that your experiment is demonstrating something that has already been shown.

Response: We appreciate your observations, and we think that after revision we are including information needed to explain the added value to the existing knowledge from our study findings. Lines 79-81: However, studies investigating the effect of Ca binding nutrients (i.e. beet pulp) in combination with the use of decreased DCAD as a preventing measure in diary ewes are lacking in literature.

- The results section needs to be shortened with emphasis on important and economical findings.

- I suggest omitting all sentences with is no significant effect observed.

Response: We very much appreciate your suggestion, but we respectfully think that results with no significance were also important in order to understand effects in our study and tried to discuss them in the last part of the revised version. Economic aspects were outside the area of interest in this study but would be an interesting future addition.

- I would suggest making every effort to explain more the results and not compare them with other studies. In general, the production data are interesting but did not discussed well.

- The discussion should include further arguments between studies or results found in the literature.

Response: We agree that this is a potential limitation of the study. We made any effort to explain briefly in accordance with the literature the results of our experiment.

Lines 280-288: Furthermore, PTH enhances the conversion of 25-dihydroxyvitamin D3 to 1.25- dihydroxyvitamin D3, that stimulates the gastrointestinal absorption of Ca [50]. A widely used acidogenic agent to decrease DCAD in livestock is ammonium chloride. Indeed, the results of our study indicated that animals fed with negative DCAD (group C, DCAD = -52 mEq/kg of DM) showed a slower decrease of Ca concentration a.p. compared with control (Group A, DCAD = +198 mEq/kg of DM). Interestingly, the same effect was noticed for animals fed sugar beet pulp without ammonium chloride (Group B, DCAD = +188 mEq/kg of DM), implying that anionic diets are not the only regulating mechanism for Ca homeostasis.

 Lines 272-282: Prevention of clinical and subclinical hypocalcemia can be achieved by activating the regulatory mechanisms of Ca homeostasis during the last days of gestation. Negative Dietary Cation Anion Difference (DCAD) between -50 and -150mEq/kg of DM during the last 10 to 28d of gestation is proposed to promote Ca homeostatic mechanisms to facilitate blood Ca concentrations within normal levels in dairy cows [8,38]. This theory is based on the development of metabolic acidosis to enhance tissue responsiveness to parathyroid hormone (PTH) [21], and thus increasing the available calcium in blood [27]. Parathyroid hormone stimulates the mobilization of Ca from the bones and decreases renal Ca excretion. Furthermore, PTH enhances the conversion of 25-dihydroxyvitamin D3 to 1.25- dihydroxyvitamin D3, that stimulates the gastrointestinal absorption of Ca [50]. A widely used acidogenic agent to decrease DCAD in livestock is ammonium chloride.

Lines 293-298: The results from the present study indicate low blood Ca levels in all groups from 4d a.p. until 12 hours p.p. (Figure 1), which is in accordance with previous studies [13]. This decrease in serum Ca concentrations was previously attributed to the reduced DMI around parturition [2,26,28] and the increased demand for Ca for colostrum formation, milk production [7,26,39], and fetal skeleton formation [36].

Lines 298-306: Interestingly, animals in Group A showed an earlier drop in Ca concentration compared to the other two groups. This leads to the assumption that the sugar beet pulp fed alone (Group B) or in combination with the acidogenic agent (Group C) contributed to a less acute drop of periparturient blood Ca concentration, probably due to enhanced utilization of food components through the improvement of palatability. This was also previously reported by Charbonneau et al., [8] in dairy cows. Age has been reported to be a major factor affecting blood Ca concentration in the periparturient period [49,58] which was in line with the results from our study (Figure 2).

Lines 315-323: Magnesium requirements raise significantly in late pregnancy and early lactation, posing the animal to the risk of subclinical or clinical hypomagnesaemia [6]. Mobilization of Mg from bone is ineffective because the ratio between Ca:Mg is 42:1 and it would disrupt Ca homeostasis [19]. Lactation and late gestation can result in large losses of Mg from intracellular and extracellular fluid, [57] which was also evident in our study, especially from 6h until 12h p.p. when udder padding with milk is commencing (Figure 3). Plasma Mg2+ is known to be influenced in a non-specific manner by catecholamines [52], insulin [48] and parathyroid hormone (PTH) [27]. The results from our research revealed no differences among nutritional groups throughout the experimental period. In contrast, Yildiz et al., [59] observed a significant decrease in Mg serum levels during the pre-partum period in ewes.

Lines 335-340: Stress of parturition is followed by an elevation of cortisol levels in blood serum which is negatively correlated with the serum levels of Ca and P, resulting in hypophosphatemia [29]. Results from the present study indicate significant differences in blood P concentrations during the periparturient period, which agrees with earlier findings [1].

Lines 347-351: Feces and milk are the most important routes of phosphate’s excretion in ruminants [22]. The ruminant mammary gland maintains constant concentrations of Ca and P in milk, independently of the availability of these elements to the maternal organism. It is possible that multiparous ewes through their higher potential in milk production, tended to withdraw increased amount of P which delayed its recovery in blood serum.

Lines 354-359: All four elements, Ca, P, Mg, and K are closely related to many metabolic events in the body. Potassium, however, is not stored to any large extent in the bone; it exists in the body primarily as a cellular constituent. As the animal ages, the readiness of availability of stored mineral elements in the bone decreases [20,30]. A high potassium concentration of the diet is a risk factor for hypomagnesemia as it reduces Mg absorption from the rumen and lowers Ca levels [54].

Lines 364-368: The determination of BHBA blood concentrations revealed no significant differences between the three nutritional groups and remained within the normal range throughout the examination period [17,31]. Blood BHBA concentration was previously reported to increase during late gestation and may reach its peak before or around lambing [15,35,51].

Round 2

Reviewer 1 Report

Accept after correcting/evaluating English by native English-speaking reviewers